# New Equivalent Thermal Conductivity Model for Size-Dependent Convection-Driven Melting of Spherically Encapsulated Phase Change Material

**DOI:** 10.3390/ma14164752

**Published:** 2021-08-23

**Authors:** Feng Hou, Shihao Cao, Hui Wang

**Affiliations:** 1College of Civil Engineering, Henan University of Technology, Zhengzhou 450001, China; 201991013@stu.haut.edu.cn (F.H.); shcao@haut.edu.cn (S.C.); 2College of Civil Engineering, Zhengzhou University of Industrial Technology, Zhengzhou 451150, China; 3School of Civil Engineering and Architecture, Hainan University, Haikou 570228, China

**Keywords:** phase change material, melting, natural convection, spherical container, equivalent thermal conductivity

## Abstract

Spherically encapsulated phase change materials (PCMs) are extensively incorporated into matrix material to form composite latent heat storage system for the purposes of saving energy, reducing PCM cost and decreasing space occupation. Although the melting of PCM sphere has been studied comprehensively by experimental and numerical methods, it is still challenging to quantitatively depict the contribution of complex natural convection (NC) to the melting process in a practically simple and acceptable way. To tackle this, a new effective thermal conductivity model is proposed in this work by focusing on the total melting time (TMT) of PCM, instead of tracking the complex evolution of solid–liquid interface. Firstly, the experiment and finite element simulation of the constrained and unconstrained meltings of paraffin sphere are conducted to provide a deep understanding of the NC-driven melting mechanism and exhibit the difference of melting process. Then the dependence of NC on the particle size and heating temperature is numerically investigated for the unconstrained melting which is closer to the real-life physics than the constrained melting. Subsequently, the contribution of NC to the TMT is approximately represented by a simple effective thermal conductivity correlation, through which the melting process of PCM is simplified to involve heat conduction only. The effectiveness of the equivalent thermal conductivity model is demonstrated by rigorous numerical analysis involving NC-driven melting. By addressing the TMT, the present correlation thoroughly avoids tracking the complex evolution of melting front and would bring great convenience to engineering applications.

## 1. Introduction

As one of promising thermal energy storage modes, latent heat storage system (LHSS) mainly absorbing/releasing a large amount of thermal energy through phase change process has received a growing interest in research community for various engineering applications, due to its significantly high energy storage per unit mass or volume [1,2]. Moreover, during phase change process, phase change material (PCM) keeps almost constant temperature, which is crucial to achieve controllable thermal performances [3]. Considering the relative high material cost and spatial occupation of directly using pure PCM as separate heat storage element, PCM is preferable to be incorporated into matrix material (e.g., cement, mortar, and concrete) to form composite LHSS in practice [4,5,6,7,8]. However, the direct incorporation of bulk PCM in matrix material may lead to poor shape-stabilizing problem, which can bring molten PCM to leak and affect the composite’s thermal performance. To overcome this obstacle, encapsulating PCM to form shape-stabilizing micro or macro particles would be a good choice in engineering applications [9,10,11].

On the other hand, it is not feasible to directly simulate the melting process of PCM, especially large amounts of PCM particles embedded in composite, because the melting process of PCM inevitably involves natural convection (NC) of liquid phase driven by temperature gradient in timely-changed liquid region, which generally requires very refined calculating mesh, tremendous simulating time, large memory capacity and complex coupled theory of heat conduction, fluid flow and moving solid–liquid interface [12]. It has been demonstrated that NC has positive effect on the total melting rate of PCM [13,14,15], but its strength is affected by heating mode, PCM shape, size, and position [16,17,18,19,20]. Hence, the contribution of NC generally cannot be ignored in the practical application of PCM and its accelerating mechanism should be discussed comprehensively.

To simplify the simulating complexity of NC-driven melting process of PCM, various strategies have been practiced by community researchers. One of these methods is introducing an equivalent pure conduction (EC) model, in which enlarged thermal conductivity of PCM is exploited to represent the contribution of NC [14,21,22,23,24], so that simpler heat conduction behavior is taken into consideration only in the equivalent model, as indicated in Figure 1a. In this context, there are a limited number of published researches available to quantitatively depict the contribution of NC in a spherical container. For example, an improved equivalent thermal conductivity (ETC) expression accounting for the NC effect for unconstrained and constrained meltings of n-octadecane enclosed in a single spherical capsule has been derived to numerically investigate the thermal performance of a whole tank consisting of 9 × 9 × 20 spherical capsules [23,24]. Similar melting fraction-based scheme with modified ETC for PCM melting was proposed by Gao et al. [25]. Different to the previous ETC approximations (see discussions in [22,25,26]) with constant characteristic length, the improved expression in a form:(1)$$\frac{k_{e}}{k_{l}}\propto {\left(T_{w}-T_{m}\right)}^{m}{\left(r_{o}-r_{i}\right)}^{n}$$
is a function of the surface temperature difference $\Delta T=T_{w}-T_{m}$ and the varied characteristic length $l_{c}=r_{o}-r_{i}$. In Equation (1), $k_{e}$ is the sought ETC, $k_{l}$ is the thermal conductivity of liquid phase, $T_{w}$ is the surface temperature of spherical capsule, $T_{m}$ is the melting temperature of PCM, $r_{i}$ is the radius of solid PCM, which is related to the time-dependent liquid fraction, and $r_{o}$ is the inner radius of spherical capsule. Besides, m and n are constant coefficients.

It should be noted that the developed ETC correlation emphasizes on capturing the melting process of PCM and are derived from the concentric sphere assumption (See Figure 1a), which is given based on the constrained melting (See Figure 1b). This means that the solid PCM will remain spherical and hold its position during the melting process. However, this is different to the real melting physics [27,28,29,30], in which the solid–liquid interface becomes non-spherical due to the nature convection of liquid PCM. Especially for the unconstrained melting (See Figure 1c), the solid PCM may drop down to contact the bottom of sphere, so the melting process is greatly different to the constrained melting. Moreover, it will cause errors to describe the shape of the non-spherical liquid region in the melting process [25]. In addition, in approximating correlation (1), the ETC is a function of time and space, thus it needs to be modified for each time step according to the molten liquid fraction to be determined in advance. This means that an iterative scheme is required and the iterative error may accumulate in depicting the melting process of PCM. Thus, correlation (1) is not easy to be used in practice.

The purpose of this study is to provide a practically simple and acceptable way to quantitatively depict the contribution of complex natural convection (NC) to the melting process. To this end, the overall energy storage efficiency for the PCM melting is addressed to reduce the difficulty in determining time-dependent characteristic length. Consequently, a new simplified empirical correlation of ETC is proposed by considering the consistency of total melting time (TMT) calculated by the NC and EC models. In practice, the TMT determines the amount of stored or released energy for PCM of given mass. Therefore, focusing on TMT and abandoning to track the complex evolution of solid–liquid interface may bring great convenience to engineers. To do this, the melting experiments of constrained and unconstrained PCMs in a spherical enclosure are firstly conducted to exhibit the difference of melting processes, and then the experiments are simulated to further explore the accelerating mechanism of NC by COMSOL Multiphysics. Next, the unconstrained melting is particularly addressed because it is closer to the real-life environment, and the effects of the PCM sphere size and the heating temperature applied on the spherical surface are numerically investigated to reveal the relation of the TMT with respect to the two critical controlling parameters, from which the accelerating effect of NC is approximately represented by enlarging thermal conductivity of PCM in an EC model. Finally, the effectiveness of the proposed empirical correlation is demonstrated by comparing the TMT calculated by the NC and EC models. This will bring great convenience in coping with a large-scale composite LHSS filled with spherical PCM capsules.

## 2. Experiment Setup

The white solid paraffin wax with a melting temperature of 298.15 K and latent heat of fusion of 212 J/g was purchased from Shanghai Joule wax company and was used as the phase change material in the work. The thermos-physical properties of paraffin wax are tabulated in Table 1 [14].

The thin-wall glass sphere with inner diameter of 62.04 mm is selected as an enclosure of PCM. The wall thickness is about 2 mm. The glass spherical enclosure has a short opening tube to facilitate the filling of the liquid PCM into the enclosure as well as the removing of air bubbles by shaking the liquid PCM during the filling process. Then, the sample is sealed with a rubber gasket and mounted on a steel supporter. The filled enclosure is placed into a refrigerator with environmental temperature 12 °C at least 24 h. The environmental temperature in the refrigerator is set to be significantly lower than the melting temperature $T_{m}=298.15\;K$ of the paraffin wax to ensure the fast solidification of the liquid paraffin. Besides, to monitor the temperature variation at the central point of the sphere, a constrained sample is prepared as well, in which a thermocouple attached to a slender wood rod is embedded at the specific position (see Figure 2).

As shown in Figure 3, the experimental system consists of a thermostatic water tank with a plastic top, a digital camera, and a temperature acquisition device (Applent AT4524 32-Channel Thermocouple Temperature Meter from Applent Instruments, Changzhou, China) connected to a desktop and the PCM container to be tested. The temperature acquisition device AT4524 has a resolution of 0.1 °C, and the k-type thermocouple connected to it has a measurement accuracy ±0.8 °C in the temperature range 0~1350 °C. First, chilly water was poured into the water tank and then was heated by an electric heating system. The setting water temperature was 318.15 K, while the actual water temperature in the tank usually had about 1 °C less than it. Next, the capsule sample to be tested was quickly taken out from the refrigerator and was then placed into the hot water to heat it. During the test, the short tube of the spherical container was mounted on the top through an opening hole. Also, the camera was used to record the transient position and shapes of the phase change interface at regular intervals during the melting process.

## 3. Computational Model

Due to the limitations of experiment, it is difficult to observe the more detailed information on fluid flow and temperature profile. Therefore, establishing a proper numerical model is necessary to extend the experimental research. Here, due to the uniform heating condition applied on the outer wall of PCM sphere and the geometric symmetry feature of the PCM sphere, a two-dimensional axisymmetric finite element model is developed using COMSOL Multiphysics software (see Figure 2) to simulate the heat transfer and melting of the PCM sphere. Three physics, including the heat conduction in the solid, the heat conduction in the liquid, and the flow driven by natural convection in the liquid, should be accounted for.

It is assumed that the flow of liquid paraffin is laminar, incompressible, and Newtonian. Also, the liquid and solid phases are assumed to be isotropic and homogeneous, and their properties keep constant in the melting process. Moreover, a mushy zone is assumed to exist along the solid–liquid interface to ensure the solving stability. In this zone, the phase change is assumed to occur in a small temperature range $\left[T_{s},\;T_{l}\right]$, where $T_{s}$ and $T_{l}$ are $T_{m}-\Delta T_{m}/2$ and $T_{m}+\Delta T_{m}/2$, respectively, and $\Delta T_{m}=T_{l}-T_{s}$ is a small temperature interval, i.e., 1 K.

Based on these assumptions, the following partial differential equation system is implemented to analyze the melting process of paraffin [14,15,31,32,33]:


*Continuity Equation:*
(2)$$\frac{\partial \rho }{\partial t}+\rho \nabla \cdot u=0$$



*Momentum Equation:*
(3)$$\rho \frac{\partial u}{\partial t}+\rho (\nabla u\cdot u)=-\nabla p+\mu \nabla^{2}u+F_{a}+F_{b}$$


*Energy Equation:*(4)$$\rho c_{p}\left(\frac{\partial T}{\partial t}+u\cdot \nabla T\right)=k\nabla^{2}T$$
where ρ is the effective density, $L_{m}$ is the latent heat of fusion, k is the effective thermal conductivity, $c_{p}$ is the effective specific heat capacity, μ is the dynamic viscosity, f is the liquid volume fraction, t is the time variable, u is the velocity vector, T is the temperature field, and p is the pressure field. Besides, $F_{a}$ is the volume force vector related to the effect of natural convection, and $F_{b}$ is the additional volume force vector which is introduced to speed up the calculational efficiency by forcing a trivial solution of u=0 in the solid phase.

Practically, when the density fluctuation caused by temperature variation is small, the flow of liquid phase can be assumed to be driven by a buoyancy force, which can be given by the following Boussinesq approximation for the NC-driven flow:(5)$$F_{a}=\rho g\beta \left(T-T_{ref}\right)$$
where $T_{ref}$ is the reference temperature, which is usually assumed to be the melting temperature $T_{m}$. β is the thermal expansion coefficient and g is the acceleration of gravity.

While, the additional term $F_{b}$ is usually given by:(6)$$F_{b}=-A(f)u$$
with Darcy assumption:(7)$$A(T)=\frac{c{\left(1-f\right)}^{2}}{f^{3}+b}$$
where $c=10^{5}$ is a big constant reflecting the morphology of the melting front and b=0.001 is a small constant used to avoid division by zero.

Obviously, in the melting process, the liquid volume fraction f is related to the temperature field T in the PCM and can be defined by:(8)$$f(T)=\{\begin{array}{l} 0,\;\;\;\;T\leq T_{s}\\ \frac{T-T_{s}}{T_{l}-T_{s}},\;T_{s}<T<T_{l}\\ 1,\;\;\;\;T\geq T_{l} \end{array}$$

Based on the definition of the liquid volume fraction, the effective thermal conductivity k, the effective density ρ and the effective specific heat capacity $C_{p}$ can respectively be given by the following mixture relations:(9)$$k=\left(1-f\right)k_{s}+fk_{l}$$
(10)$$\rho =\left(1-f\right)\rho_{s}+f\rho_{l}$$
(11)$$C_{p}=\frac{1}{\rho }\left[\left(1-f\right)\rho_{s}C_{p,\;s}+f\rho_{l}C_{p,\;l}\right]+L_{m}D(T)$$
where the subscript s represents the solid phase and the subscript l represents the liquid phase. D(T) is the standard Gaussian function which is zero everywhere except the temperature interval ΔT. More importantly, its integral is equal to 1.

Moreover, the solid phase can be regarded as a liquid with very high viscosity, so the dynamic viscosity μ in Equation (3) can be describe by the parameter A(T) in the simulation:(12)$$\mu =\mu_{l}\left(1+A(T)\right)$$

In addition, the temperature boundary condition is applied on the outer surface of the container. The heat conduction only takes place along the thickness of the glass wall of the container. The thermal conductivity of glass is set as 0.81 W/m/K, which is chosen from COMSOL Material Library.

In the finite element simulation coupling the fluid flow and the heat conduction in solid, an extremely refined mesh with triangular axisymmetric element is used to obtain the grid-independent convergence solution. Besides, the boundary layer technology is used near the common interface between the PCM and the glass inner wall, and the number of boundary layers are set as 7, as shown in Figure 4. The total number of elements are 14,036. In addition, the computing time step is set as 0.1 s in the simulation. The direct solver PARDISO in COMSOL is used.

## 4. Results and Discussion

In this section, the melting experiments are simulated to further explain the difference of melting process of constrained and unconstrained PCM and explore the melting mechanism driven by the natural convection. In the simulation, the initial temperature is $T_{0}=285.15\;K$, and the heating temperature applied on the surface of the spherical container is $T_{w}=318.15\;K$. Then, a detailed computational analysis is performed for the unconstrained melting, which is closer to the real-life physics, and a correlation of equivalent thermal conductivity is proposed to simplify the analysis.

### 4.1. Validation of the Numerical Model

#### 4.1.1. Constrained Melting

Figure 5 shows the transient shapes of the solid paraffin within a spherical container with inner diameter of 64.04 mm during the constrained melting process. The heating temperature is set as $T_{w}=318.15\;K$. It is seen that the solid paraffin holds its position under the constraint of the thermocouple and the slender wood rod, but its volume gradually shrinks with the evolution of melting. Initially, the solid PCM is wrapped in a very thin liquid film, indicating that the melting of paraffin is dominated by heat conduction between the inner surface of the container and the solid paraffin at this stage. As the melting of the solid PCM continues, the paraffin in the upper part melts faster than that in the lower part, and as a result, the solid–liquid interface becomes non-spherical and the solid PCM becomes smaller and smaller. Simultaneously, the portion of liquid PCM occupying the top region is always higher than that at the bottom region. This may be attributed to the lateral whirl flow caused by NC of liquid paraffin. At the final stage, the remaining solid PCM drops from the constrained position and contacts the bottom of the container until it fully melts. Besides, during the melting process, the bottom surface of the solid PCM is observed to become bumpy (see Figure 6). This may because of the scouring action of bottom whirlpool flow caused by the rising up of the warmer liquid from the bottom and the sink of the cooler liquid. As a whole, the lateral NC leads to higher rate of melting than the bottom NC. As a comparison, Figure 5 also displayed the simulated melting fronts in the paraffin at different time instances. The blue and red colors represent the solid and liquid phases, respectively, and the black arrow represents the velocity direction. It is found from Figure 5 that the evolution of the simulated melting front is basically consistent with the experimental results, although the melting time in the experiment is slightly longer than that in the simulation. This can be attributed to the fact that the actual heating temperature in the experiment is slightly lower than the desired one, and the thermal resistance between the glass wall and the PCM, the glass wall and the surrounding water is ignored in the computation.

Further, Figure 7 shows the variation of computational temperature at the center point A of the spherical container for the constrained melting case (see Figure 2). From Figure 7, it is observed that the melting process around the center point A is roughly divided into three stages: solid heating (stage 1), phase transition (stage 2), and liquid heating (stage 3). At stage 1, the PCM around point A keeps the solid state, so heat is transferred to it in the form of heat conduction. Gradually, as the temperature exceeds the melting point, the solid PCM melts and becomes liquid phase. At this stage, the temperature almost keeps constant. Finally, at stage 3, the heat conduction and flow in the liquid PCM dominates the temperature at point A to rise fast and reach the heating temperature at the end. Besides, it should be observed from Figure 7 that the time needed for the melting of PCM at point A in the numerical simulation is slightly shorter than that in the experiment. The reason may be the relatively lower actual heating temperature and the presence of interfacial thermal resistance in the experiment. In spite of this, the experimental result shows a similar trend to the simulated result.

To further explain the NC effect on the melting of solid PCM in the spherical container, the distributions of velocity and temperature at 70 min are plotted in Figure 8, from which the thermally stable and unstable regions of the liquid PCM are clearly observed, and the separating line between them is close to the top of the mushy zone. In the stable region, the thermal energy transfer mainly happens by heat conduction. However, it is dominated by NC in the three unstable regions. In the two big unstable regions, the hot liquid moves upwards along the inner wall of the glass sphere and the cold liquid moves downwards along the melting front. Therefore, anticlockwise whirlpool flows respectively form in the two regions, in which the melting of solid PCM accelerates. Moreover, the bottom whirlpool flow makes the bottom of solid PCM become bumpy, while the lateral whirlpool flow makes the solid–liquid interface of PCM become inclined. Additionally, there is a small unstable region between the two big unstable regions, in which a clockwise whirlpool flow takes place. As a result, the melting front becomes more and more irregular as the melting goes on.

Besides, to compare the accelerating effect of NC on the melting of paraffin, the model without NC, that is the pure heat conduction (PC) only, is simulated and the melting front is displayed in Figure 9, which indicates that the melting front keeps concentric spherical shape in the melting process under the heat conduction behavior transferring from outwards to inwards, and its size gradually decreases as the melting goes on. This is typically different to the NC model. Moreover, the total melting time (TMT) required for completing the paraffin melting is 205 min, which is significantly longer than 113 min when the NC is considered.

#### 4.1.2. Unconstrained Melting

For the case of unconstrained melting, no temperature data can be directly recorded in the experiment, so only the shape and location of the melting front are observed. Contrary to the constrained melting, the unconstrained melting in the spherical container exhibits different experimental phenomenon [20], as displayed in Figure 10. At the beginning, the paraffin melts to form a thin liquid layer along the interior surface of the spherical shell. Then, the solid PCM sinks to contact the bottom of the container and the liquid PCM rises up to form lateral whirlpool flow, due to the higher density of the solid paraffin than the liquid paraffin. Besides, because of the contact of the solid PCM with the bottom of container, the heat conduction is dominant near the bottom region of the container and has a higher melting efficiency than the NC. As a result, the unconstrained melting shows a faster melting speed than the constrained melting.

Besides, Figure 10 shows the variation of the melting front at specific time instances obtained from the computational model. It can be seen that the computational profiles of the melting front are highly similar to the experimental profiles. The thermal transfer in the upper region of the PCM sphere is mainly NC, which causes the melting front inclined. Moreover, due to the density difference between the liquid and solid PCM, the solid PCM gradually moves downwards and contacts the bottom of the container as the melting goes on. Therefore, the heat conduction takes place at the bottom of solid PCM. Moreover, the downward movement of the sold PCM pushes the warm liquid layer between the solid PCM and the inner wall of the sphere to rise along the lateral wall of the sphere. This enhances the strength of lateral whirl flow and accelerates the melting of the upper region of the PCM sphere. Additionally, the total melting time is about 100 min for the experiment and 85 min for the computational model, respectively. The TMT difference of the experiment and the numerical simulation may be attributed to the ignored thermal resistance between the glass wall and the PCM in the simulation and the relatively lower actual heating temperature in the experiment.

### 4.2. Size Effect of PCM Sphere on Natural Convection

Next, the size effect of PCM sphere on NC is investigated using the verified numerical model for the unconstrained melting case, which is closer to the real-life applications. The wall thickness of the spherical container is 1 mm, and the sphere size is adjusted by the inner radius R changing from 30 to 2 mm. The solid wall material is steel with thermal conductivity 44.5 W/m/K from COMSOL Material Library. Hereafter, the steel wall is used because the macro-encapsulated PCM sphere by a steel shell is particularly used for potential energy building applications. The steel shell can provide both sufficient protection to the PCM core and excellent thermal and mechanical properties of the PCM sphere [11,34]. A heating temperature $T_{w}$ is uniformly applied on the outer surface of steel sphere. $\Delta T=T_{w}-T_{m}$ is the temperature difference of the heating temperature $T_{w}$ and the melting temperature $T_{m}$ of the PCM.

Figure 11 displays the variation of liquid fraction against the heating time for the case of the temperature difference ΔT=10 K. It can be seen from Figure 11 that the liquid fraction increases as the melting goes on. Moreover, as the size of PCM sphere decreases, the needed TMT decreases as well. As comparison, the variation of liquid fraction for the PC model is plotted in Figure 11 as well. It is observed that the TMT of the PC model obviously becomes longer than that of the NC model, and the difference of them becomes more significant as the sphere becomes larger. However, it is interesting that the PC and NC models give almost the same TMT when R≤3 mm. This means that the melting of the PCM in a small sphere is mainly dominated by heat conduction, instead of NC, whose strength is suppressed in a small cavity.

To clearly illustrate the variation of TMT, various ΔT and R are considered and the results are listed in Table 2, from which it is observed that the TMT for the NC model ($t_{NC}$) is always less than that for the PC model ($t_{PC}$), owing to the accelerating effect of NC on the melting of PCM. Moreover, both $t_{NC}$ and $t_{PC}$ change in terms of the spherical inner radius R and the temperate difference ΔT. Generally, the smaller the sphere size, and/or the higher the heating temperature, the shorter the TMT. Interestingly, it is found that the TMT ratio $t_{PC}/t_{NC}$ basically descends with the decrease of sphere size and reaches a relative stable value independent of the heating temperature when R is 2 mm, as displayed in Figure 12. This indicates that the NC effect is size-dependent. It becomes weaker for the smaller size of sphere and can be ignored in practice when R=2 mm or less.

### 4.3. Equivalent Thermal Conductivity Model

From the results above, it is confirmed that the size-dependent NC plays an import role in the melting process of PCM. However, it is not feasible to take the complex NC effect into account in the practical engineering applications, especially for composite LHSS including amounts of encapsulated PCM particles, because the simulation procedure of NC involves complex coupled theory and inevitably leads to tremendous computing cost of time and memory capacity. To simplify this procedure, in this work, an equivalent pure conduction (EC) model is established by counting the contribution of NC by an enlarged thermal conductivity of PCM. In the EC model, only the energy equation involving only heat conduction is considered for the phase change simulation, so a lot of computing cost can be saved.

In order to distinguish the inherent thermal conductivity $k_{PCM}$ of PCM, the enlarged thermal conductivity in the EC model is named as equivalent thermal conductivity (ETC) $k_{e}$. Because of the fact that the higher the thermal conductivity of PCM, the shorter the TMT, it is reasonable to assume that the TMT of PCM and the ETC in the EC model basically meets the following approximating inverse proportional relation
(13)$$\frac{k_{e}}{k_{PCM}}\times \frac{t_{NC}}{t_{PC}}=1.038$$
from which $k_{e}/k_{PCM}$ can be evaluated by the results of $t_{PC}/t_{NC}$ listed in Table 2 for different heating temperatures and sphere sizes. It is noting that the coefficient 1.061 is set to ensure the TMT calculated by the target value of $k_{e}$ in the EC model is closest to that in the NC model.

In the practical application, the ETC $k_{e}$ can be expressed as a function of R and ΔT. Generally, the approximate expression of $k_{e}$ can be written as [33]:(14)$$\frac{k_{e}}{k_{PCM}}=C\cdot {\mathrm{Ra}}^{m}=C\cdot {\left(\mathrm{Gr}\cdot \mathrm{Pr}\right)}^{m}=D\cdot \Delta T^{m}\cdot R^{n}$$
where:(15)Ra=Pr⋅Gr
is Rayleigh number:(16)$$\mathrm{Gr}=\frac{g\beta \Delta TR^{3}}{\mu_{l}{}^{2}}$$
stands for Grashof number, and:(17)$$\mathrm{Pr}=\frac{\mu_{l}\rho_{l}c_{p}}{k_{l}}$$
denotes Prandtl number.

The data fitting of the target value of $k_{e}$ and the corresponding parameters ΔT and *R* gives: (18)D=52.9, m=0.1706, n=0.6837

To check the accuracy of the proposed approximate correlation (14) with the fitting coefficients (18), the TMT of PCM in the EC model with the new $k_{e}$ given in Equation (14) is evaluated for the case of ΔT=10 K and is compared to that from the NC model. Results in Table 3 indicate that the EC model can be used to effectively treat the melting problem of PCM sphere with great convenience.

It is worth pointing out that the correlation (14) is given under the condition of constant heating temperature on the surface of sphere. For composite LHSS including amounts of encapsulated PCM microspheres, the surface temperature of each microsphere can be assumed to be uniform due to the relatively small size of microsphere compared to the composite bulk, and can be evaluated by averaging the surface temperature of each microsphere.

## 5. Conclusions

In this paper, the constrained and unconstrained melting in a glass spherical container is firstly investigated by experiments and numerical simulation to reveal the difference of melting process of PCM and also to validate the axisymmetric computational model. The different melting mechanisms caused by NC-driven flow are clearly demonstrated in the melting process of PCM. Then, the NC effect on heat transfer in the paraffin PCM encapsulated by a steel spherical shell is analyzed under various sphere sizes and heating temperatures. the dependence of NC on the sphere size and the heating temperature for unconstrained melting is investigated through numerical simulation and a simple correlation on the ETC of PCM is presented. The conclusions of this study include:(1)For the constrained melting, the heat transfer is controlled by heat conduction across the glass wall at the early stage of the melting process, and then, as the melting goes on, the NC strengthens and the lateral whirl flow makes the top region of the PCM sphere melt faster than the bottom region. Moreover, the wavy profile at the bottom of solid PCM is formed due to the action of the bottom whirl flow.(2)For the unconstrained melting, the solid PCM gradually sinks to contact the bottom of the inner glass wall and thus the melting at the bottom is dominated by heat conduction across the inner wall. While the melting of the top region of the solid PCM speeds up due to the lateral whirl flow strengthened by the downward movement of the solid PCM.(3)The size of the PCM capsule has an important influence on the NC-driven heat transfer in the melting process. With the decrease of the capsule’s radius, the difference between the TMT of the PC and NC models is gradually narrowing. Specially, when the radius of the PCM capsule is less than 2 mm, the NC effect is suppressed significantly in the spherical cavity and the heat transfer is basically controlled by heat conduction.(4)Generally, the high heating temperature applied on the surface of spherical shell shortens the TMT of PCM. However, when its influence can be ignored when the sphere radius is less than 2 mm.(5)In order to simplify the complex calculation of the melting of the PCM sphere, the contribution of the NC effect is converted into an enlarged thermal conductivity of the PCM, and a correlation is proposed by keeping the TMT unchanged.

## Figures and Tables

**Figure 1 materials-14-04752-f001:**
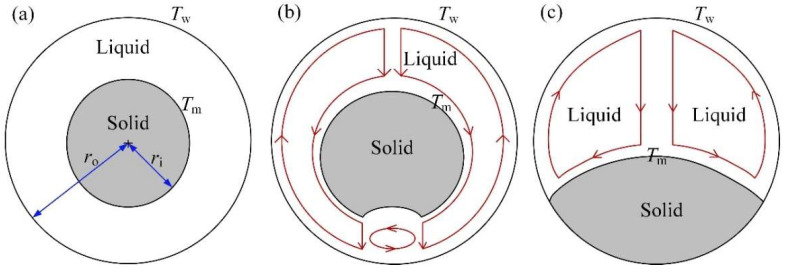
Schematic diagrams of: (**a**) equivalent thermal conductivity model, (**b**) constrained melting, and (**c**) unconstrained melting.

**Figure 2 materials-14-04752-f002:**
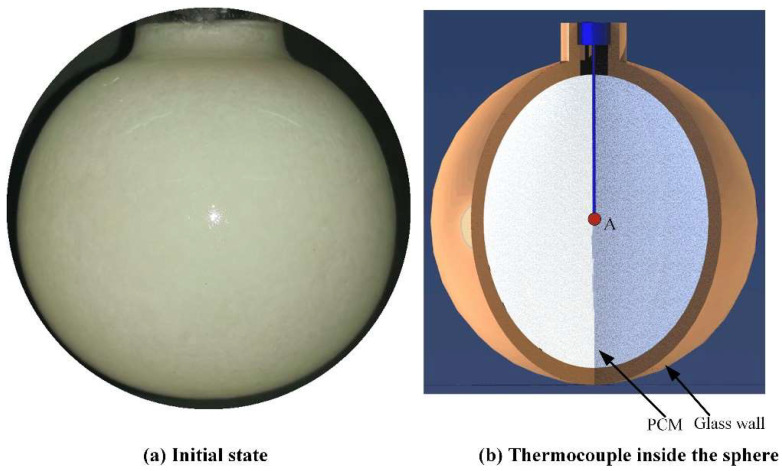
(**a**) The spherical enclosure filled with paraffin wax and (**b**) the schematic diagram of the constrained computational domain embedding with a thermocouple located at the center of the sphere.

**Figure 3 materials-14-04752-f003:**
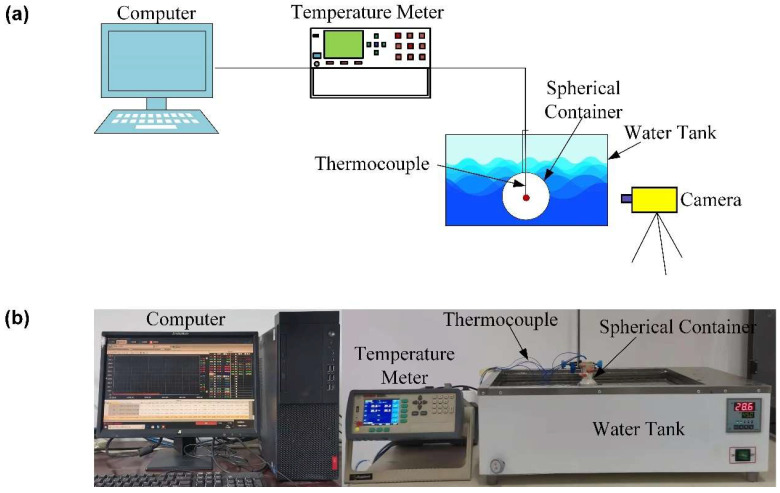
(**a**) Schematic diagram and (**b**) actual picture of the experimental setup for the constrained sample.

**Figure 4 materials-14-04752-f004:**
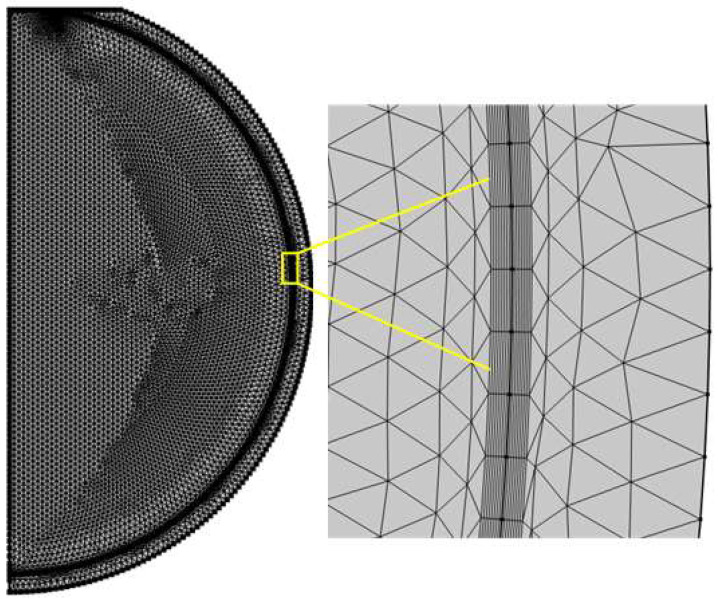
The mesh grid used in the two-dimensional axisymmetric finite element model.

**Figure 5 materials-14-04752-f005:**
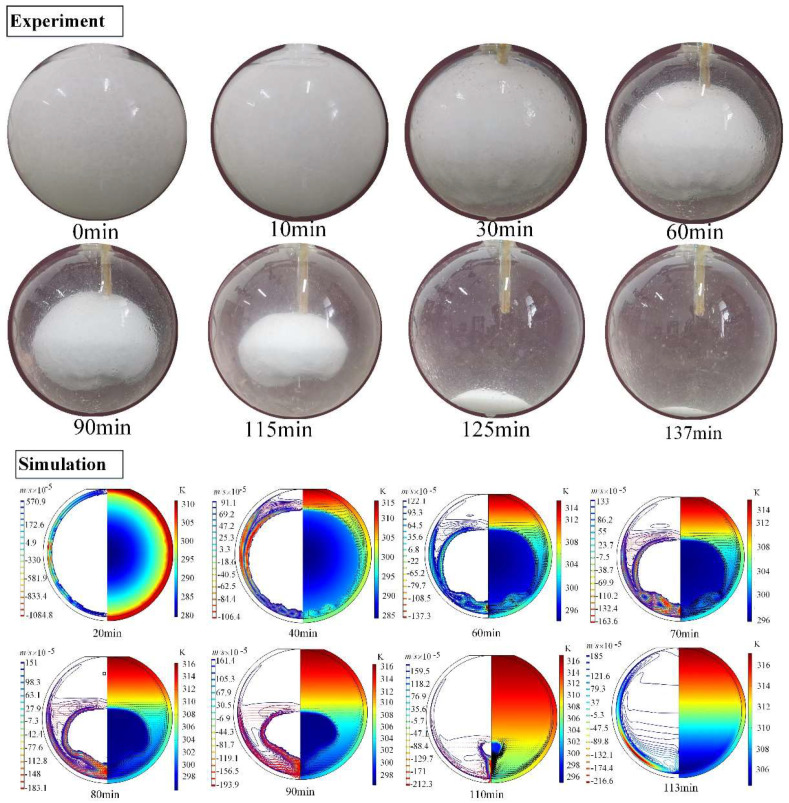
Transient shapes of the solid paraffin for the constrained melting inside the spherical container with $R=31.02\;\mathrm{mm}\;\mathrm{and}\;\;T_{w}=318.15\;K$.

**Figure 6 materials-14-04752-f006:**
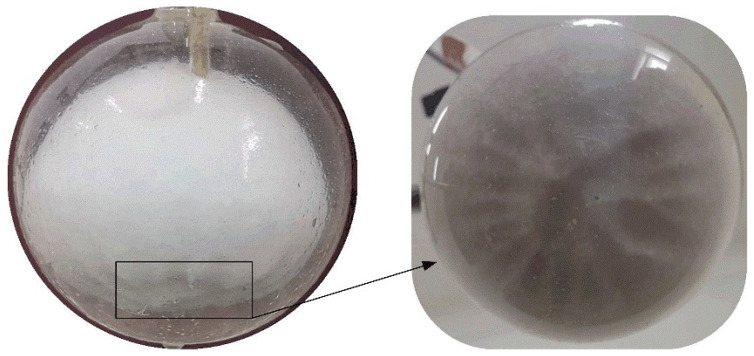
Bottom waviness profile of the sold PCM at 60 min for the constrained melting inside the spherical container with $R=31.02\;\mathrm{mm}\;\mathrm{and}\;\;T_{w}=318.15\;K$.

**Figure 7 materials-14-04752-f007:**
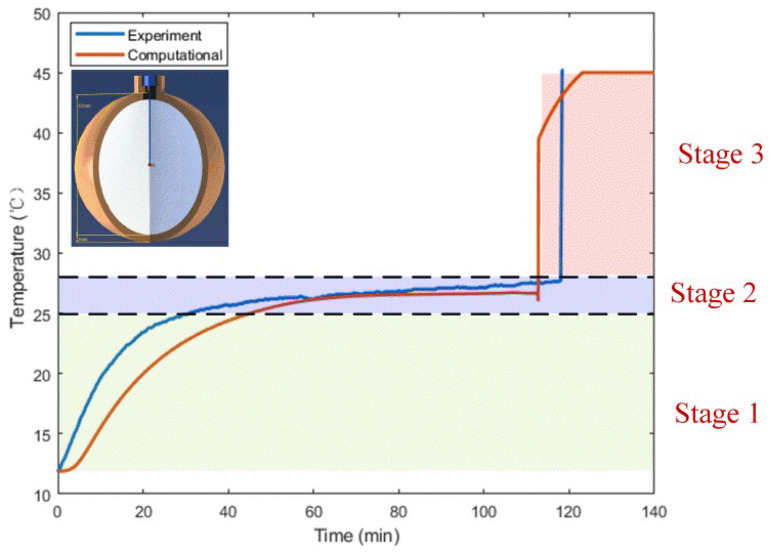
Comparison of the computational and measured temperatures at the central point of the spherical container during the constrained melting process with $R=31.02\;\mathrm{mm}\;\mathrm{and}\;\;T_{w}=318.15\;K$.

**Figure 8 materials-14-04752-f008:**
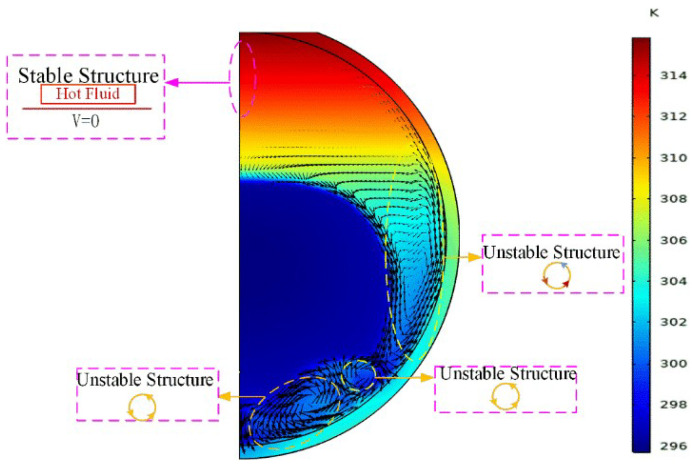
Thermally stable and unstable regions for the constrained melting at 70 min inside the spherical container with $R=31.02\;\mathrm{mm}\;\mathrm{and}\;\;T_{w}=318.15\;K$.

**Figure 9 materials-14-04752-f009:**
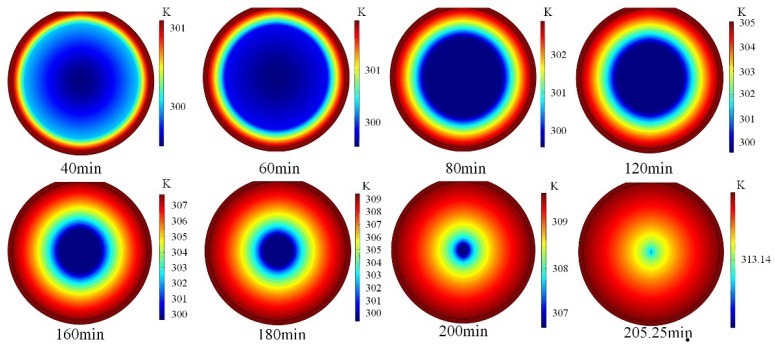
The PCM melting process without natural convection for the constrained melting inside the spherical container with $R=31.02\;\mathrm{mm}\;\mathrm{and}\;\;T_{w}=318.15\;K$.

**Figure 10 materials-14-04752-f010:**
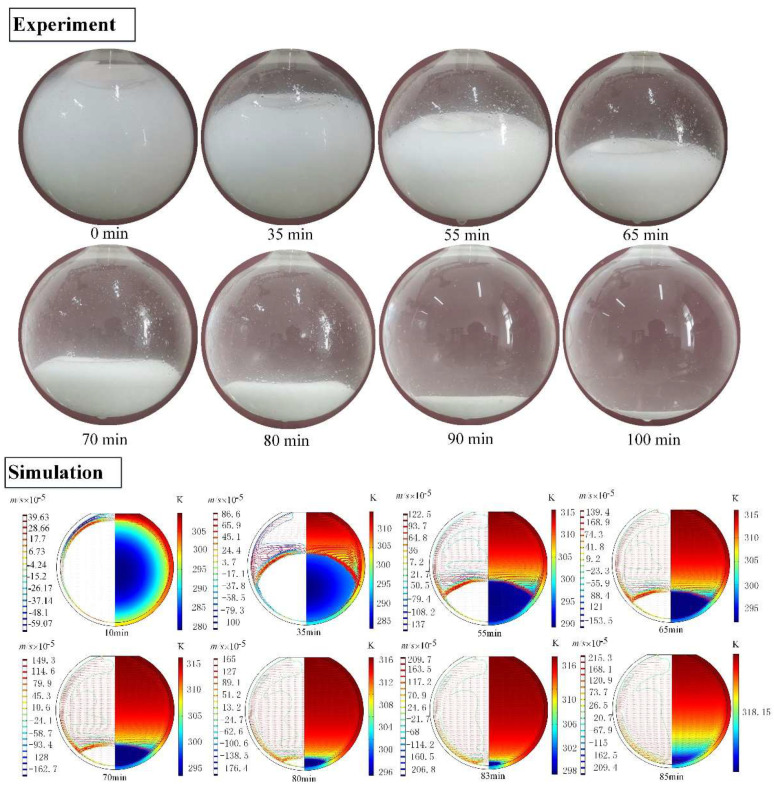
Transient shapes of the solid paraffin for the unconstrained melting process inside the spherical container with $R=31.02\;\mathrm{mm}\;\mathrm{and}\;T_{w}=318.15\;K$.

**Figure 11 materials-14-04752-f011:**
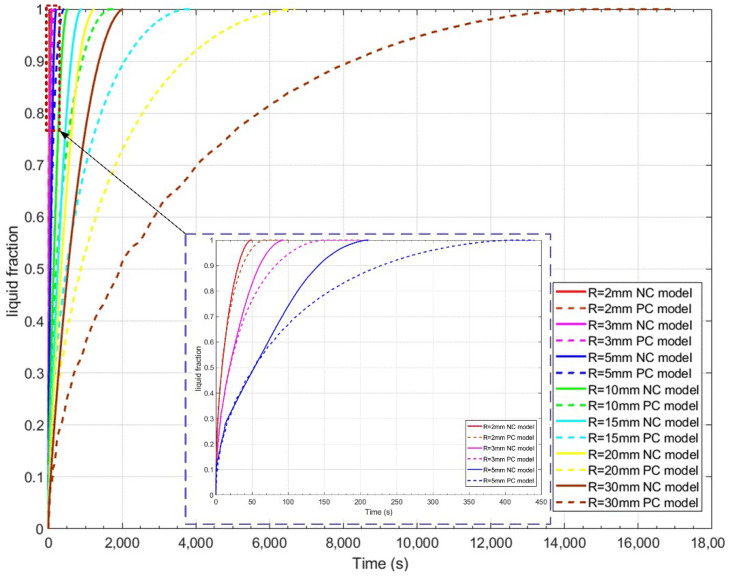
Comparison of liquid fraction calculated by the PC and NC models for the case of ΔT=10 K.

**Figure 12 materials-14-04752-f012:**
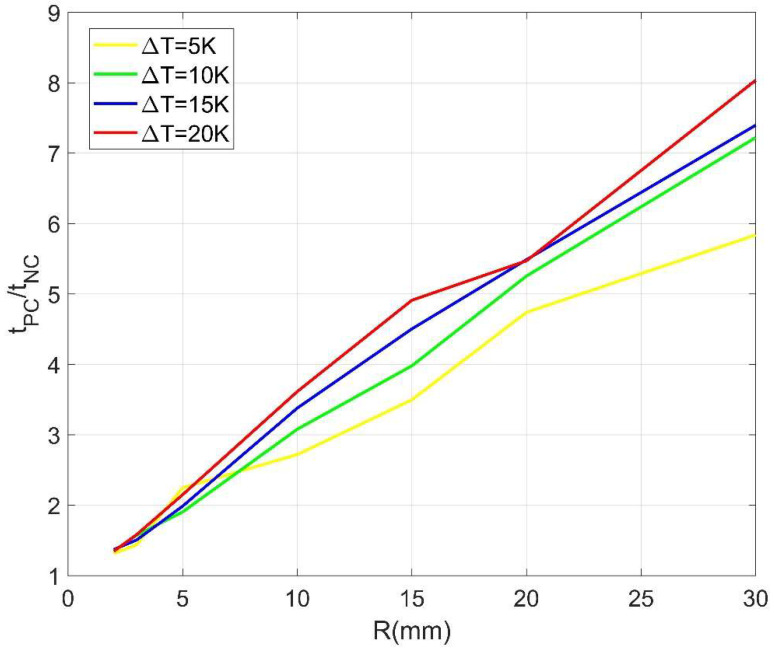
Variations of TMT ratio for different sphere sizes and heating temperatures.

**Table 1 materials-14-04752-t001:** Thermo-physical properties of paraffin wax [14].

Property	Value
Melting temperature $T_{m}$ (K)	298.15
Solid density $\rho_{s}$ ($\mathrm{kg}/m^{3}$)	849.7
Liquid desity $\rho_{l}$ ($\mathrm{kg}/m^{3}$)	814.8
Dynamic viscosity $\mu_{l}$ (Pa·s)	0.00579
Solid specific heat $c_{ps}$ (J/(kg·K))	2400
Liquid specific heat $c_{pl}$ (J/(kg·K))	3220
Solid thermal conductivity $k_{s}$ (W/(m·K))	0.2
Liquid thermal conductivity $k_{l}$ (W/(m·K))	0.15
Latent heat of fusion L (J/g)	212
Thermal expansion coefficient β ($K^{-1}$)	0.001

**Table 2 materials-14-04752-t002:** Total melting times for the NC and PC models with different sphere sizes and heating temperatures.

ΔT (K)	R (mm)	$t_{NC}\;\left(s\right)$	$t_{PC}\;\left(s\right)$	$t_{PC}/t_{NC}$
5	30	4638	27,075	5.8376
	20	2544	12,054	4.7382
	15	1939	6778	3.4956
	10	1087	2954	2.7176
	5	330	748	2.2667
	3	189.6	272.7	1.4383
	2	88.3	116	1.3137
10	30	2020	14,585	7.2203
	20	1247	6551	5.2534
	15	911	3624	3.9780
	10	532	1637	3.0771
	5	210.7	401	1.9032
	3	93.5	147	1.5722
	2	48	65	1.3542
15	30	1422	10,517	7.3959
	20	844.9	4635	5.4859
	15	573	2580	4.5026
	10	331	1118	3.3776
	5	133	264	1.9850
	3	63.6	95.8	1.5063
	2	34.1	46.7	1.3695
20	30	1036	8324	8.0347
	20	671.8	3675	5.4703
	15	421	2067	4.9097
	10	250	903	3.6120
	5	100.1	215	2.1479
	3	47.4	75	1.5823
	2	25.3	34	1.3439

**Table 3 materials-14-04752-t003:** Total melting times for the NC and EC models for different sphere sizes and ΔT=10 K.

ΔT (K)	R (mm)	$t_{NC}\;\left(s\right)$	$t_{EC}\;\left(s\right)$	Error
10	30	2020	1956	3.2%
	20	1247	1196	4.1%
	15	911	888	2.5%
	10	532	511	3.9%
	5	210	206	1.9%
	3	94	91	3.2%
	2	48	47	2.1%

## Data Availability

The data presented in this study are available on request from the corresponding author.

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
