# Peer review of "New Equivalent Thermal Conductivity Model for Size-Dependent Convection-Driven Melting of Spherically Encapsulated Phase Change Material"

_materials, 2021, doi:10.3390/ma14164752_

Round 1

Reviewer 1 Report

The paper deals with the constrained and unconstrained melting of a phase change material inside a spherically encapsulated geometry, and It is very well written. I am recommending publication with minor comments:  

Please try to use better-referencing software.  I found many places that reference number is missing :

  • In line 61 please refer to the reference, rather than referring to the name of the group.
  • In Eq.(1) the equal sign is missing.
  • Line 71 reference number is missing.
  • Line 90 reference number is missing.
  • Line 94 reference number is missing
  • Line 409 reference number is missing
  • Line 411 reference number is missing
  • Line 418 reference number is missing

-       What you mean by dense suspension layer. 
-       What is the difference between Eq.(2) and Eq.(3)?
-       Move the Caption of Figure 2 to the same page as its Figure. 
-       If it is possible to write the equations with a professional math editor, they are difficult to read sometimes.

Reviewer 2 Report

Dear Authors,

Here are my comments and suggestions:

  1. Lines 154-192 contain results rather than experimental details. My suggestion is to modify the Figures so that you present the melting process real images next to the simulated ones. In this way one can have a clear image of the results. The manuscript is difficult to follow if you need to go three pages up to find the images and compare them to the simulations.
  2. Figures 9-12 containing the simulations do not contain any bars/color scales, what are the temperatures? Velocities?
  3. In general, all figure captions are laconic, more details are required. Figures along with captions should be self-explanatory, contain experimental details or other relevant information on what they represent.
  4. Figure 8. “Comparison of the computed and measured temperatures at the central point A.”, does not support the comment in Line 278: “the experimental result shows good agreement with the simulated result on the whole.”- The two graphs are quite different.
  5. Figure 13 contains a lot of data that cannot be visualized, an overlaid zoom window can be a solution.
  6. Figure 14- error bars missing
  7. Lines 425-427: “the NC effect on heat transfer in the paraffin PCM encapsulated by a glass spherical shell is analyzed under various sphere sizes and heating temperatures.”, but in Line 341-346 the sphere solid wall is described as steel.
  8. The conclusions of this study are rather intuitive: for sure the radius of the capsule has a great influence on the heat transfer process and for smaller radius, heat conduction becomes dominant.
  9. Lines 23-24, The statement “The effectiveness of the effective thermal conductivity model is demonstrated by rigorous numerical analysis involving NC-driven melting” is not supported by the presented results (lines 380-425). Authors define an equivalent thermal conductivity, in eq (13) and then in lines 408-414 refer to results in Table 3 that ”indicate that the EC model can be used to effectively treat the melting problem of PCM sphere with great convenience.”
  10. Please correct the error messages in the text.

General remarks: the idea of the study is interesting, but it still needs some work. The manuscript is rather difficult to follow as it starts with experiments and then for each simulation recalls previous figures and so on; the last part needs to be consolidated to validate the “new equivalent thermal conductivity model”.

Reviewer 3 Report

The paper deals with the study experimentally and numerically the process of melting of paraffin wax as phase-change material (PCM) in spherical geometry. The wax fusion has been studied experimentally in completely filled sphere of 62 mm of interior diameter for constrained and unconstrained samples by measuring the melting time and the temperature at its center point when in bath temperature is 45ºC. These experiments are simulated for equivalent thermal conductivity model, constrained melting and unconstrained melting by COMSOL Multiphysics. A simplified empirical correlation of effective thermal conductivity is proposed from total melting time calculated by computational models.

The paper could be accepted for publication, after some major changes.

The authors should broad the description of the experimental equipment used. They should add the resolution, interval and accuracy of the measured properties. The experimental equipment used to measure the temperature and control in the water bath are poor acurate.

In table 2 and in the text, it is recommended that the temperature be expressed in Kelvin, the density in kg/m3. It is desirable more significant figures in the thermal conductivity and thermal coefficient expansion of the material. For a better thermophysical characterization of the material, the specific heat and thermal conductivity values in liquid and solid phases should be indicated.

In equation 1, typographical error in the sign "=" must be corrected.

In lines 71, 90, 94, 242, 409, 410, 411 and 418 shows "Error! Reference source not found.", It should be corrected.

In 4.1 Validation of numerical model, figure 9 shows an appreciable difference between the experimental results and those obtained from the simulation compared to the statement of lines 278 and 279. It should be explained or the computational model improved. In the same way, the different results of the total melting time, 100 minutes versus 85 minutes, should be considered.

In 4.2 Size Effect of PCM sphere on natural convection, Figure 14 shows anomalous values in the results obtained for the 20 mm radius spheres at the temperature differences 5 and 20ºC. A revision of the calculations at this point would be advisable.

In 4.3 Equivalent thermal conductivity model, mean value for the set of data of the deviation between the results of the correlation and that obtained from the comparison of the NC and EC models, could be of interest for determining the accuracy of the proposed correlation, not only the error in the variation of the radius for ΔT = 10ºC.

The literature review must be updated include references to similar works.  The following references are of interest.

- Ettouney, H. El-Dessouky, A. Al-Ali. Heat Transfer During Phase Change of Paraffin Wax Stored in Spherical Shells. J. Sol. Energy Eng. 2005, 127(3), 357-365. https://doi.org/10.1115/1.1850487

- Assis, L. Katsman, G. Ziskind, R. Letan. Numerical and experimental study of melting in a spherical shell. International Journal of Heat and Mass Transfer 2007, 50, 1790–1804. doi:10.1016/j.ijheatmasstransfer.2006.10.007

Round 2

Reviewer 2 Report

Thank you for adressing all the comments.